# Pseudomonassin, a New Bioactive Ribosomally Synthesised and Post-Translationally Modified Peptide from *Pseudomonas* sp. SST3

**DOI:** 10.3390/microorganisms11102563

**Published:** 2023-10-15

**Authors:** Kevin Jace Miranda, Saif Jaber, Dana Atoum, Subha Arjunan, Rainer Ebel, Marcel Jaspars, RuAngelie Edrada-Ebel

**Affiliations:** 1Marine Biodiscovery Centre, Department of Chemistry, University of Aberdeen, Meston Walk, Aberdeen AB24 3UE, UK; subha.arjunan@syngenta.com (S.A.); r.ebel@abdn.ac.uk (R.E.); m.jaspars@abdn.ac.uk (M.J.); 2College of Pharmacy and Graduate School, Adamson University, 900 San Marcelino Street, Ermita, Manila 1000, Philippines; 3Strathclyde Institute of Pharmacy and Biomedical Sciences, University of Strathclyde, John Arbuthnott Building, 161 Cathedral Street, Glasgow G4 0RE, UK; saif.jaber@strath.ac.uk (S.J.); ruangelie.edrada-ebel@strath.ac.uk (R.E.-E.); 4Department of Pharmaceutical Chemistry, Faculty of Pharmaceutical Sciences, The Hashemite University, Zarqa 13133, Jordan; dana.atoum@hu.edu.jo

**Keywords:** genome mining, metabolomics, ribosomally synthesised and post-translationally modified peptide, *Pseudomonas* sp. SST3

## Abstract

Genome mining and metabolomics have become valuable tools in natural products research to evaluate and identify potential new chemistry from bacteria. In the search for new compounds from the deep-sea organism, *Pseudomonas* sp. SST3, from the South Shetland Trough, Antarctica, a co-cultivation with a second deep-sea *Pseudomonas zhaodongensis* SST2, was undertaken to isolate pseudomonassin, a ribosomally synthesised and post-translationally modified peptide (RiPP) that belongs to a class of RiPP called lasso peptides. Pseudomonassin was identified using a genome-mining approach and isolated by means of mass spectrometric guided isolation. Extensive metabolomics analysis of the co-cultivation of *Pseudomonas* sp. SST3 and *P. zhaodongensis* SST2, *Pseudomonas* sp. SST3 and *Escherichia coli,* and *P. zhaodongensis* SST2 and *E. coli* were performed using principal component analysis (PCA) and orthogonal projections to latent structures discriminant analysis (OPLS-DA), which revealed potential new metabolites in the outlier regions of the co-cultivation, with other metabolites identified previously from other species of *Pseudomonas.* The sequence of pseudomonassin was completely deduced using high collision dissociation tandem mass spectrometry (HCD-MS/MS). Preliminary studies on its activity against the pathogenic *P. aeruginosa* and its biofilm formation have been assessed and produced a minimum inhibitory concentration (MIC) of 63 μg/mL and 28 μg/mL, respectively.

## 1. Introduction

Genome mining in natural products research has accelerated the discovery and isolation of new and novel natural products. It has become the core approach for natural products discovery due to the identification of biosynthetic gene clusters (BGCs) that encode for a diverse arrays of secondary metabolites, from polyketide synthases (PKS) and non-ribosomal peptide synthases (NRPS) to ribosomally synthesised post-translationally modified peptides (RiPPs) [1,2]. Genome sequences of a diverse range of organisms such as bacteria, cyanobacteria, and fungi have become more and more accessible through the use of public-domain gene sequence repositories and reduced sequencing costs, which have enabled the investigation of BGCs and hence of the specialised metabolites they encoded [3]. The encoded BGCs in the genome of an organism involved in the biosynthesis of specialised metabolites are often silent or cryptic under laboratory conditions. This often poses difficulties in targeting and isolating new compounds with potential bioactivity [4]. A technique that is commonly used to elicit the production of secondary metabolites and the activation of some gene clusters in bacteria is co-cultivating it with other organisms. This could be another bacterium or fungus to enhance the production, target metabolites, express the silent genes, or produce new secondary metabolites from their interactions. This approach also shows features that are not produced in the single-strain culture, and it often leads to the elucidation of new metabolic pathways [5]. Most of the co-culture experiments that have revealed new biosynthetic pathways and new secondary metabolites are between bacteria and fungi due to their competition for nutrients and fermentation conditions that differ between the strains in the co-culture experiment [6]. Bioinformatics allows us to map the metabolome of organisms from their gene clusters. Several approaches dealing with the optimisation, evaluation, and isolation of bioactive secondary metabolites using metabolomics platforms and algorithms have been successfully employed using different bioinformatics tools [7,8,9]. Since metabolites are considered to be the final product of entire cellular processes, the outcomes of enzymatic processes are also observed in the study of the metabolome of an organism. Thus, metabolomics became useful in drug discovery, discovering biomarkers for disease, as well as in the study of plants, bacteria, and fungi [10,11].

Lasso peptides form a class of RiPPs that have a lariat knot topology in which the tail is threaded through the macrolactam ring [12]. Lasso peptides are classified according to the disulphide bridges in their topology: Class I has two disulphide bridges; Class II, the most common type of lasso peptide. has no disulphide bridges; finally, Class III and Class IV each have one disulphide bridge; the Class III has a disulphide bridge that interlinks the C-terminus of the tail and the N-terminus of ring and Class IV has a disulphide bridge on the tail [13]. The lasso topology of these peptides confers stability to their structures and makes them resistant to proteases and thermal degradation [13,14]. Even though this type of RiPP is structurally (in terms of the disulphide bridges they have) and functionally diverse, they still follow a collective biosynthesis in which they are all encoded in the gene as a precursor peptide, which consists of the N-terminal leader sequence and the C-terminal core sequence [15]. RiPPs have been investigated as a new source of antimicrobial therapeutics because of their specific activity against pathogens; for example, the antibiotic microcin J25 from *E. coli* which is only active against *Salmonella* and *Shigella* spp., which both belong to the Enterobacteriaceae family as *E. coli* [16,17,18]. Their stability in the face of both thermal and proteolytic degradation also made them promising therapeutics [19,20].

This work aims to elicit and characterise the new RiPP **1** that has been identified using genome mining and bioinformatics tools, which was identified as a lasso peptide. The elicitation was carried out by co-cultivating two *Pseudomonas* spp. isolated from the same environment in South Shetland Trench. We also performed dereplication and metabolomics analysis of the extracts using PCA and OPLS-DA and revealed potential new metabolites, alongside with known tryptophan- or phenylalanine-containing phenolate siderophores that have been also identified in the genome. The entire sequence of the RiPP **1** predicted from the BGC was confirmed using stepwise HCD-MS/MS, which enables us to fragment all amino acid residues of the peptide, making this technique an invaluable tool in RiPPs research.

## 2. Materials and Methods

### 2.1. Organism and Fermentation

Deep-sea bacteria *P. zhaodongensis* SST2 and *Pseudomonas* sp. SST3, were collected as part of the PharmaDeep Project in South Shetland Trench (SST), Antarctica [21], and the *E. coli* K-12 strain was obtained from the laboratory of Dr. Hai Deng. Small-scale cultures of modified GYM medium containing 4.0 g yeast extract, 10.0 g malt extract, 4.0 g glucose, 12.0 g of calcium carbonate, distilled water up to 1 L, and pH 7.20 were prepared by inoculating 50 mL of media with a single colony of the organism and incubating it for 7 days at 28 °C with shaking at 150 rpm. For the monoculture, 50 mL of the medium was used. For the co-cultivation, 50 and 25 mL of each medium were used for each organism. The prepared co-cultivation flasks were as follows: 50 mL *Pseudomonas* sp. SST3 and 50 mL *P. zhaodongensis* SST2, 50 mL *Pseudomonas* sp. SST3 and 50 mL *E. coli*, 50 mL *P. zhaodongensis* SST2 and 50 mL *E. coli*, 50 mL *Pseudomonas* sp. SST3 and 25 mL *P. zhaodongensis* SST2, and 50 mL *P. zhaodongensis* SST2 and 25 mL *Pseudomonas* sp. SST3. The co-cultivation flasks were combined on the 6th day and shaken for 24 h prior to extraction. Large-scale fermentation was conducted on 9 L of medium in which 500 mL of the culture of *P. zhaodongensis* SST2 culture was placed in twelve 2 L flasks and 250 mL of *Pseudomonas* sp. SST3 culture was placed in twelve 1 L flasks and combined together on the 6th day of incubation.

### 2.2. Extraction and Isolation

Diaion HP-20 (Fisher Scientific UK, Loughborough, UK) (3 g/50 mL) were placed in each flask to absorb the secondary metabolites produced on the 6th day and shaken for 24 h. Culture broths with HP-20 were then filtered using glass wool (Sigma-Aldrich, St. Louis, MO, USA), washed with water to remove excess salts, and then extracted with methanol (100 mL × 3). Successive methanol extracts were combined and concentrated under reduced pressure. Fractionation was conducted using solid-phase extraction (SPE) chromatography C-18 column (Phenomenex Strata) with 25%, 50%, 75%, and 100% methanol (Fisher Scientific UK). Final purification was conducted using C-18 reverse phase high-performance liquid chromatography (HPLC) (Agilent, Santa Clara, CA, USA) with 35–100% 95:5 water:methanol (solvent A) and 100% methanol (solvent B) gradient for 35 min.

### 2.3. HCD-MS/MS Analysis

High-resolution HCD-MS analyses were conducted at the Institute of Medical Sciences, University of Aberdeen, Forester Hill Campus, using a Q Exactive Plus hybrid quadrupole—Orbitrap mass spectrometer (Thermo Fisher Scientific, Waltham, MA, USA). The sample in 0.1% formic acid was infused directly from the syringe pump at 5 µL/min to the HESI source. Instrument settings for MS/MS of 871.46 *m*/*z* ion (z = 2). Fragmentation NCE was varied stepwise from 20 to 50. MS/MS was also acquired over a lower scan range (50.0 to 750.0 *m*/*z*) for some NCE values. Acquisition time was from 0.2 min to 0.5 min. Direct infusion positive-mode HCD was applied over the range of 20–50 NCE. In the Thermo^®^ Orbitrap instrument, NCE is linearly correlated with the mass-to-charge (*m*/*z*) of the precursor ion. This set-up of the instrument allows for the acquisition of a high-resolution fragmentation, particularly of peptides below 200 Daltons (Da) and over 2000 Da mass range, and the data are automatically gathered regardless of the mass of the analyte [22].

### 2.4. Metabolomics Analysis

Multivariate analysis (MVA) was conducted using SIMCA 15.0.1 (Umetrics, Umeå, Sweden) software, and we set the parameters for the principal component analysis (PCA) and orthogonal projections to latent structures discriminant analysis (OPLS-DA) algorithm. We converted the positive-mode MS data of the extracts using MZMine and Proteowizard. Data processing from MZMine followed a step-by-step protocol from peak detection, deconvolution, deisotoping, filtering, alignment, gap filling, the addition of adducts and complexes for eliminating misassignments of predicted molecular formulae in the algorithm, and the threshold of molecular formula was set at 5 ppm. The processed MZMine data were then converted into a CSV file and input at SIMCA to generate score plots. These score plots measure the similarities and differences of the metabolites in the mass spectrometry data. To further discriminate and characterise the data of the extracts, we go into the PCA and OPLS-DA analysis and generated graphs.

### 2.5. Antibacterial Assay

Minimum inhibitory concentration (MIC) was conducted using alamarBlue (Invitrogen, Waltham, MA, USA) broth dilution assay [23,24]. The test organism, *P. aeruginosa* (ATCC 27853), was prepared by inoculating a loop of each bacterium in 5 mL of Luria-Bertani (LB) broth (GIBCO). The test organism was incubated for 16 h at 37 °C to reach the stationary phase. After 16 h, 100 µL of aliquot was transferred to new culture containing 5 mL LB broth. The new culture was incubated for 6 h at 37 °C to reach the log phase. The assay was performed in 96-well plates (Thermo Fisher UK) by mixing 10 µL of prepared extracts, gentamicin as the positive control, and dimethylsulfoxide (DMSO) (Fisher Scientific UK) as the negative control in the dilution plate with 90 µL of LB broth containing bacteria to give a final concentration of 100 µg/mL, and this was incubated at 37 °C for 16 h. A 10 µL of 10% alamarBlue solution was added to each well and incubated for another 4 h. Absorbance readings at the 560 nm excitation wavelength and the 590 nm emission wavelength (Hidex) were conducted to evaluate cell viability [24].

### 2.6. Planktonic Assay Solution for Biofim Inhibition

Assay plates were prepared in the same manner as the cell viability test with 10 µL of the extract, gentamicin, and 90 µL of LB broth containing bacteria, which was incubated for 16 h at 37 °C. After incubation, the wells were emptied and washed twice with 100 µL of phosphate buffer saline (PBS). Absorbance readings at 600 nm were undertaken to determine biofilm formation viability and activity by means of turbidity [25,26].

## 3. Results

### 3.1. Genome Mining and Prediction of the Structure of Pseusomonassim

A deep-sea bacterium was obtained from South Shetland Trench (SST), Antarctica, as part of the PharmaDeep Project [21]. Whole-genome sequencing revealed that the species belongs to the *Pseudomonas* genus, *Pseudomonas* sp. SST3. Genome mining studies using antibiotics and Secondary Metabolite Analysis Shell (antiSMASH) [27], a bioinformatics tool that annotates and compares gene clusters from known sequences that encodes for secondary metaboliutes, showed the annotated lasso peptide biosynthetic gene clusters (BGCs) consisting of the precursor peptide (pdnA), an ATP-dependent protease similar to cysteine/transglutamase (pdnB), an ATP-dependent macrolactam synthetase with homology to glutamine (Gln) synthase (pdnC), and the ABC transporter (pdnD). The prediction of the structure of the lasso peptide relies on the sequence of the precursor peptide in the gene cluster. The precursor peptide allows for the identification of the target sequence of the mature peptide and its confirmation using mass spectrometry (MS) [11]. In our search for the putative structure of this interesting peptide using a genome mining approach, we entered the sequence of the precursor peptide of this gene cluster into BLAST [28] and translated the sequence to the RiPPMiner [29] platform, which gave three possible putative structures (Appendix A). Since these kinds of peptides have a motif governing the number of amino acid residues in both ring and tail [30], the structure with the most plausible number of residues was chosen (Figure 1). The predicted peptide has an accurate mass of 1740.9060 Da.

### 3.2. Elicitation of the Target RiPP

A single culture of *Pseudomonas* sp. SST3 and *P. zhaodongensis* SST2 and the co-cultivation of *Pseudomonas* sp. SST3 with *P. zhaodongensis* SST2, *Pseudomonas* sp. SST3 with *E. coli*, and *P. zhaodongensis* SST2 with *E. coli* were used for the target RiPP in modified GYM medium. We evaluated which of the co-cultivations or the monoculture was the best for the large-scale production of the target RiPP. In order to obtain sufficient yields of the peptide **1** for potential isolation and full structural characterisation, approaches to stimulate increased production of the target compound by the two *Pseudomonas* strains, SST2 and SST3, co-occurring in the SST, and both containing the respective BCGs for the lasso peptide, were evaluated. Thus, we developed a co-cultivation protocol of the two strains, *P. zhaodongensis* SST2 and *Pseudomonas* sp. SST3, with each other, as well as the terrestrial Gamma-proteobacterium *E. coli*, and also including the relevant controls (Appendix A). All cultures were fermented in modified GYM medium as stated above. The rationale behind this approach was that these types of RiPPs are known to have a narrow spectrum of antibacterial activity within the same class. An example for this finding is microcin J25, isolated from *E. coli*, which displayed activity against *E. coli* H157:O7, *Shigella*, and *Salmonella*, all of which belong to the Enterobacteriaceae family [31,32]. In addition to the single culture of *Pseudomonas* sp. SST3 as described above, expression of the target peptide **1** was detected in the co-cultivation of *P. zhaodongensis* SST2 and Pseudomonas sp. SST3 and the co-cultivation of *Pseudomonas* sp. SST3 and *E. coli*. Conversely, no production of the peptide was observed in the single culture of *P. zhaodongensis* SST2 or in its co-culture with *E. coli*. A semiquantitative assessment of the titres was obtained by comparing the relative intensity of the base peak at *m*/*z* of 871.45, corresponding to the doubly charged ion [M+2H]^2+^ of the predicted peptide **1**, and through a metabolomics approach using SIMCA, multivariate analysis (MVA), such as principal component analysis (PCA), and orthogonal projections to latent structures discriminant analysis (OPLS-DA).

### 3.3. Pseudomonassin Characterisation Using High-Energy Collision Dissociation Tandem Mass Spectrometry (HDC-MS/MS)

The peptide in the genome of *Pseudomonas* sp. SST3 was elicited in the co-cultivation medium. The sequence of the entire peptide was elucidated using high-energy collision dissociation tandem mass spectrometry (HDC-MS/MS). The precursor mass [M+2H]^2+^ 871.45 +/− 0.02 *m*/*z* of the peptide was selected in all HCD analyses. In this experiment, we observed that there is little or almost no fragmentation of product ions observed at 20 normalised collision energy (NCE) for the doubly charged peptide species, and the precursor ion [M+2H]^2+^ 871.45 +/− 0.02 *m*/*z* dominates the spectrum. At this point, the higher-mass fragments from the tail residues are of the *b*-ions (b_12_, b_14_, b_15_, and b_16_); these ions have a mass above 1000 Da and some of the y-ions (y_5_ and y_6_), both oh which have less than 1000 Da in mass also appear on the spectrum. As the energy increases, the precursor ion is fragmented into product ions more effectively, and more fragments from the lower mass region are observed [33]. At 30 NCE, this energy level was able to show the mass of all the fragments of the peptide; thus, we deduced its entire sequence from tail down to the macrolactam ring and the location of the ring opening. Here, we find the 30 NCE as the optimum collision energy, wherein we observed both the b- and y-ion fragments, respectively (Appendix A). The opening of the macrolactam ring was observed to be at the site of the glutamic acid sequentially going to glycine, as we mined from the putative structure. The precursor mass of the peptide is still observable at a high intensity, and the product ions of the peptide, from highest to lowest molecular weight of each amino acid component of the b-ions, are seen in the spectrum. For the y-ions, y_11_, y_15_, and y_16_ cannot be observed at this collision energy. As we increased the NCE to 40 and 50, the shifting of the fragmentations from higher masses to lower ones were already appearing [34,35,36], the precursor ion [M+2H]^2+^ 871.45 +/− 0.02 *m*/*z* was no longer observable, and we started to see the smaller products’ ions. At this collision energy, the spectrum became highly dominated by ions less than 400 Da, and the lowest possible fragmentation of the amino acids were seen in the spectrum on both b- and y-ions. (Appendix A). The HCD-MS/MS method was preferred for the complete elucidation and sequencing of pseudomonassin.

ProteinProspector [37] readily calculates the theoretical accurate masses expected upon fragmentation of the ring, can be compared to those observed at 30% collision energy, and displayed all the b- and y- ions of the fragmented peptide in 30 NCE (Table 1).

### 3.4. Metabolomics Analysis Reveals Potential New Metabolites from the Co-Cultivation of Two Pseudomonas Species

We utilised two multivariate analysis (MVA) methods, principal component analysis (PCA) and orthogonal partial least squares discriminant analysis (OPLS-DA), to extract information from the data gathered in the LC-MS on the extracts of the monoculture and co-cultivation of *Pseudomonas* sp. SST3, *P. zhaodongensis* SST2, and *E. coli*. The total number of *m*/*z* features from the monoculture and co-cultivation extracts is 30,059. These features were processed in the mzMine to remove the noise and background peaks of the culture medium. From these mass features, we conducted the dereplication of the extracts using a Microsoft Excel macro and coupled them with information from the database Dictionary of Natural Products (DNP). The principle of this approach, which, in part, is based on proprietary protocols established in the group of Dr. RuAngelie Edrada-Ebel, is described in a series of recent publications [38,39,40,41]. The hits were filtered according to the bacterial source set to the genus *Pseudomonas*, and 22 potential hits were identified on the basis of matching molecular formulae. Some of these compounds are produced by *Pseudomonas fluorescens* and include the family of pseudomonic acid, from which the commercial topical antibiotic mupirocin is derived [42,43]. This dereplication is a good indication of the diversity of the metabolites being produced by deep-sea organisms in both monoculture and co-cultivation settings (Appendix A).

Prior to multivariate analysis (MVA), the mass spectral data were pre-processed in mzMine to separate the signals from the noise [44]. MVA was initiated with PCA to give an overview of the observations (samples) in correlation with their respective variables (features). In this study, observations represented the samples from the different co-cultivation parameters and ratios of the organisms employed, while the variables were the observed LC-MS ion peaks for various metabolites. The PCA of eight extracts gave R2 (goodness of fit) and Q2 (predictability) values of 0.923 and 0.881, respectively. Good predictability and fit values indicated a linear correlation between the co-cultivation extracts and the production of the metabolites. In general, the R2 should not exceed Q2 by more than 3 units to ensure that a model is not over-fitted [45]. In the PCA scores plot (Figure 2A), the two *Pseudomonas* spp. co-cultivated with *E. coli* were separated from the cluster (labeled as KM490 and KM492). The cluster consisted of the single and co-cultures of two *Pseudomonas* spp. for the loadings plot (Figure 2); the target peptide was found amongst the discriminating metabolites on the third quadrant (lower left) corresponding to that of KM491. Dereplication of discriminating metabolites in the PCA loadings plot revealed compounds of the amonabactin family, a class of tryptophan or phenylalanine-containing siderophores, namely, amonabactin P693 *m*/*z* 693.3010 [M+H]^+^, amonabactin P750 *m*/*z* 750.3217 [M+H]^+^ (4.39), and amonabactin P789 *m*/*z* 789.3335 [M+H]^+^, respectively. These dereplicated compounds are of NRPS origin and have been annotated in the *Pseudomonas* sp. SST3 genome. Using DNP, NP Atlas, [46], and Reaxys [47] databases gave no hits for ion peaks at *m*/*z* [M+H]^+^ 674.3524, 694.3082, 758.4085, and 1124.6382, whereas ion peaks at *m*/*z* [M+H] 530.2971, 576.3515, 652.4073, 671.4139, and 751.3670 afforded hits originating from plants, fungi, and Gram-positive microorganisms such as Streptomyces and *Micromonospora* species, which were disregarded. The unidentified discriminating metabolites for KM490 and KM492 could potentially be new metabolites and warrant further investigation for targeted isolation in future.

An OPLS-DA of the mass spectral data was also accomplished. It was employed to construct a model for ease of interpretation of the discriminating spectral features (metabolites) responsible for the separation of the assigned groups. In this case, either the monoculture or the co-culture extracts were considered to determine the production of the target peptide. Interpretation of the OPLS-DA scores and loadings plots was the same as the PCA [44]. For the OPLS-DA scores plot (Figure 2), the sample extracts were grouped according to the type of culturing approach, as in monocultures (right quadrants) versus co-cultures (left quadrants). The co-cultures were further grouped according to the employed organisms. In the OPLS-DA loadings plots, discriminating feature *m*/*z* [M+H]^+^ 318.1129 could be assigned to the monocultures, while ion peaks at *m*/*z* [M+H]+ 598.4905 and [M+2H]^2+^ 871.4596 were assigned for the co-cultures. However, due to the separation of KM492 from the rest of the other co-culture extracts, the variation within the groups (57.4%) was greater than that between groups (11.7%). Such separation further indicated that the discriminating ion peak at *m*/*z* [M+H]^+^ 598.4905 was unique to KM492. The ion peak at *m*/*z* [M+2H]^2+^ 871.4596 was dereplicated as the doubly charged target peptide, which correlated with the co-culture extracts with *E. coli*.

### 3.5. Antibacterial and Biofilm Assay

Pseudomonassin was tested against the pathogenic *P. aeruginosa* with gentamicin as the standard, as well as for the biofilm formation. The method that has been used for the biofilm inhibition is the planktonic assay method. This test is used to assess the bacterial attachment using the staining method for the biomass. It shows moderate activity against the pathogen with an MIC of 63 μg/mL and a good activity against biofilm formation at 28 μg/mL (Figure 3).

## 4. Discussion

The elicitation of the peptide and comparison with the LC-MS/MS revealed that the co-culture of *P. zhaodongensis* SST2 and *Pseudomonas* sp. SST3, in a 2:1 volume ratio of the culture broths, gave the highest intensity, i.e., twice as much compared to the 1:1 volume ratio, 3 times higher when compared to the co-culture with *E. coli*, and up to 40 times higher when compared with the monoculture of *Pseudomonas* sp. SST3 (Appendix A). Obviously, due to the differences in ionization efficiency in LC-MS/MS, there is no intention to imply that the intensity of the respective ion accurately reflects the concentration of the target compound in each extract; nonetheless, this analysis served to optimise fermentation conditions for the subsequent isolation of **1** upon large-scale fermentation.

In the OPLS-DA analysis, potential new metabolites also arising from the co-culture have been identified. For the ion peaks *m*/*z* [M+H]^+^ 318.1129 and 598.4905, no hits were found from any of the databases used, suggesting that these discriminating features can be potentially new metabolites. Using a metabolomics approach on chemical profiling studies assisted the prioritisation of the co-cultivation method and predicted the probability of each of the conditions to elicit the production of the target peptide as well as other potential new metabolites. The metabolomics approach also lowers the chance of the redundancy of re-isolating known metabolites for a more efficient targeted isolation work. This is the first time that this kind of peptide isolated from its native strain has shown activity against this type of pathogen and inhibition of biofilm formation. Continuous research and trials on how to combat biofilm formation in clinical isolates of *P. aeruginosa* is still a difficult task due to the mutations that lead, once again, to resistance against several drug combinations. These findings on its bioactivity, as well as its ability to inhibit biofilm formation, present the potential of this class of RiPPs as a therapeutic model or drug delivery method against selective pathogens.

## 5. Conclusions

Using a co-cultivation technique, we have successfully elicited the class of RiPPs called lasso peptides in *Pseudomonas* sp. SST3, pseudomonassin. LC-MS analysis highlighted that the co-culture of *P. zhaodongensis* SST2 and *Pseudomonas* sp. SST3 gave the highest intensity of the target peptide. Subsequent fragmentations of the *b*- and *y*-ions of the tail confirmed the correct sequence, as predicted by RiPPMiner. The metabolomics analysis revealed several discriminating metabolites between the single and the co-cultures, some of which are assumed to belong to the amonabactin family of siderophores, as suggested by their molecular formulae. Furthermore, several potentially new compounds, with no hits in natural products databases, represent promising targets for future isolation and structural characterization.

## Figures and Tables

**Figure 1 microorganisms-11-02563-f001:**
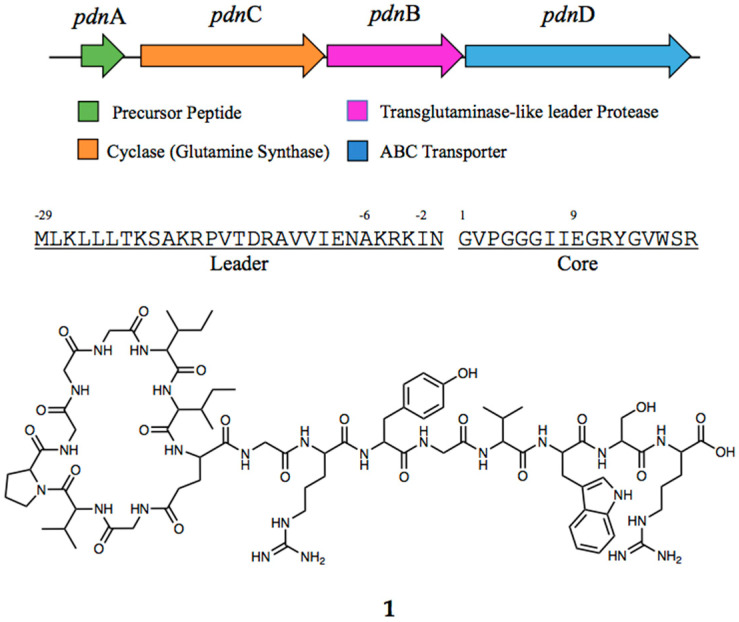
BGC of pseudomonassin (**1**) showing the four main gene clusters for its biosynthesis and the sequence of the leader and core peptide.

**Figure 2 microorganisms-11-02563-f002:**
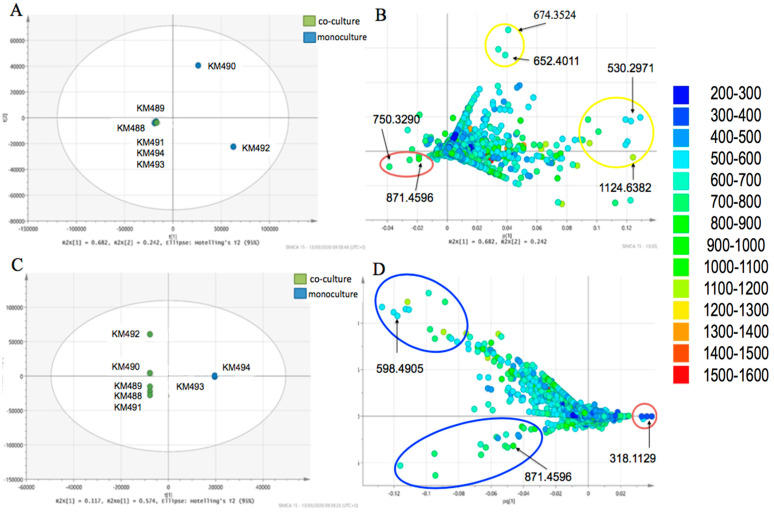
PCA scores plot (**A**) and loading plots (**B**) and OPLS-DA scores plot (**C**) and loading plots (**D**) of the single culture and co-cultivation of *Pseudomonas zhaodongensis* SST2, *Pseudomonas* sp. SST3, and *Escherichia coli* Highlighted in blue, red and yellow circles are the featured *m/z* outliers, which could be potential new metabolites and the double charged *m/z* for pseudomonassin.

**Figure 3 microorganisms-11-02563-f003:**
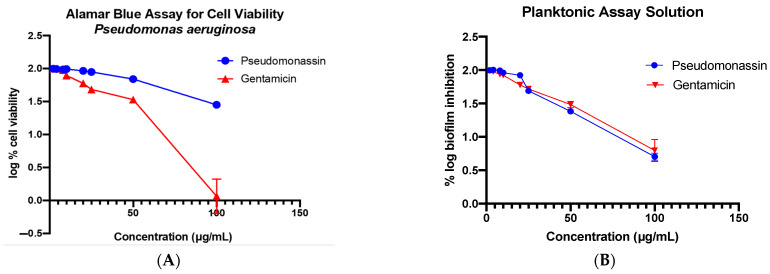
Bioassay results graph showing the activity of pseudomonassin against Pseudomonas aeruginosa (**A**) and the inhibition of biofilm formation (**B**).

**Table 1 microorganisms-11-02563-t001:** Fragmentation pattern observed on the *a-*, *b-,* and *y*-ions: the loss of water from the N-terminus and C-terminus from ProteinProspector and the actual mass fragments from 30 NCE HCD-MS/MS.

Amino Acid	*b-ions*			*y-ions*		
	Theoretical MW Fragment	Observed MW Fragment	Error	Theoretical MW Fragment	Observed MW Fragment	Error
G	-	-	-			
V	157.0972	157.0971	0.64 ppm	1684.8918	-	-
P	254.1499	254.1495	1.57 ppm	1585.8234	-	-
G	311.1714	311.1710	1.29 ppm	1488.7706	1488.7701	−0.34 ppm
G	368.1928	368.1923	1.36 ppm	1431.7492	1431.7497	0.34 ppm
G	425.2143	425.2137	1.41 ppm	1374.7277	1374.7292	1.09 ppm
I	538.2984	538.2979	0.93 ppm	1317.7062	-	
I	651.3824	651.3818	0.92 ppm	1204.6222	1204.6205	−1.41 ppm
E	762.4145	762.4138	0.92 ppm	1091.5381	1091.5361	−1.83 ppm
G	819.4359	819.4351	0.98 ppm	980.5061	980.5026	−3.57 ppm
R	975.5370	975.5354	1.64 ppm	923.4846	923.4833	−1.41 ppm
Y	1138.6004	1138.5991	1.14 ppm	767.3835	767.3825	−1.30 ppm
G	1195.6218	1195.6205	1.09 ppm	604.3202	604.3201	−0.17 ppm
V	1294.6908	1294.6897	0.85 ppm	547.2987	547.2984	−0.55 ppm
W	1480.7696	1480.7712	−1.08 ppm	448.2303	448.23	−0.67 ppm
S	1567.8016	1567.8020	−0.26 ppm	262.151	262.1507	−1.14 ppm
R	-	-	-	175.1190	175.1188	−1.14 ppm

## Data Availability

Whole genome sequence of *Pseudomonas* sp. SST3 is publicly available in GenBank.

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
