# Peer review of "Pseudomonassin, a New Bioactive Ribosomally Synthesised and Post-Translationally Modified Peptide from Pseudomonas sp. SST3"

_microorganisms, 2023, doi:10.3390/microorganisms11102563_

Round 1
Reviewer 1 Report
Dear Authors,
After reviewing the manuscript entitled "Pseudomonassin, a new bioactive ribosomally synthesized and post-translationally modified peptide from Pseudomonas sp. SST3", I have decided to suggest that it be accepted for publication in Microorganisms Journal with minor changes, as I think that the manuscript is well written, the structure and results are well presented. Also, the characterization of a new peptide with biological activity is important as part of the alternatives being investigated, due to bacterial resistance to antibiotics. However, the manuscript has some minor details that the authors should correct, particularly in the discussion.
Kind regards,

Reviewer 2 Report
This manuscript provides some interesting results using modern, technical approaches. I have only a few comments.
Line 156: The final data collection on biofilm inhibition is not clear. No biofilm staining is mentioned. Another approach would be to measure biofilm by production of planktonic cells after washing and incubation, but no further incubation is mentioned. The reporting of the results only adds to the confusion. Please clarify.
I am really surprised that the organisms being studied can be grown at 37 C, given that they were isolated from Antarctic ocean sediment and lab grown at 10 C in the isolation process. While growth range per se isn't important for the results of this paper, the discrepancy between source and lab growth conditions is striking and deserves comment.
Line 317: I am unfamiliar with log % or % log, and they are not typically used in literature with which I am familiar. Is a value of 2 equal to 100% (i.e 10^2)? Could these data be shown in a more familiar way, e.g. using just percent? I would make a recommendation for Fig. 3B, but I still do not understand the assay procedure. If the results are stained, planktonic cells originating from surviving biofilm, then a simple % inhibition from zero down to 100% would make more sense (or alternatively, 100% and "percent of control biofilm"). If Fig. 3B measures the presence of planktonic cells by OD, it would be a suspension, not a solution.
Line 322: I would have expected a table or figure showing these culture combinations and results, despite the disclaimer that technical reasons prevent the amounts from being particularly accurate. The effect of co-culturing with related organisms is of particular interest despite there being similar observations already published.
The English is quite good, and I only noticed a couple of odd things in passsing. Another check of the language is always helpful.
Reviewer 3 Report
The manuscript by K.J. Miranda and coworkers deals with the extraction and mass characterization of a peptide from a co-cultivation of two deep-sea bacteria. The manuscript reads well, still a couple of paragraphs would benefit from a revision by a mother-tongue English scientist. The results are interesting and worth publishing in microorganisms. Minor points are reported below:
Par 2.2. Line 107: please, explain abbreviations when first encountered in the text (e.g., HP-20?; “with methanol (MeOH)”; Line 111-112: please, doublecheck eluant A, it is likely the other way round (95:5 water/methanol)
P.4, Par. 3.1. The text seems to report on earlier work, already published. If so, it should be moved in the introduction section.
P. 5, Par. 3.2, Lines 181-184: “Monoculture and co-cultivation […] in modified GYM medium.” (?) statement not clear, please rephrase it. Line 185-186: “for potential isolation” (?) why do the authors need huge amounts of peptide 1 for its isolation? It might be that they aim at obtaining huge quantities of the crude mixture from which peptide 1 can be extracted. Please, rephrase the sentence.
P.5, Par 3.3. Please check English language with the help of a mothertongue English scientist (e.g., “some of the y-ions (y5 and y6), both are below 1000 Da appearing...” ?, “this energy level were able to identify…” etc.).
Typos:
Abstract, line 17-18: “[…] to isolate pseudomonassin, a class of ribosomally-“; “modified peptides (RiPP)”; “analysis […] was”. By the way, since later in the text peptide 1 is called “pseudomonassin”, could the class of peptides possibly be “pseudomonassins”?
Paragraphs of section 3 “Results” should be renumbered to read “3.1”, “3.2”, etc.
Par. 3.1. (to be), line 160: “bacterium was obtained”
All names of microorganisms should be in italics (e.g., par. 3.2(to be): “Pseudomonas sp.”, etc.)
P.8, line 311 “as well as for its ability to hamper biofilm formation”
The manuscript would benefit from a revision by a mother-tongue English scientist.
